# The SNP rs2298383 Reduces ADORA2A Gene Transcription and Positively Associates with Cytokine Production by Peripheral Blood Mononuclear Cells in Patients with Multiple Chemical Sensitivity

**DOI:** 10.3390/ijms21051858

**Published:** 2020-03-09

**Authors:** Attilio Cannata, Chiara De Luca, Liudmila G. Korkina, Nadia Ferlazzo, Riccardo Ientile, Monica Currò, Giulia Andolina, Daniela Caccamo

**Affiliations:** 1Department of Biomedical Sciences, Dental Sciences and Morpho-functional Imaging, Polyclinic Hospital University, Messina 989125, Italy; attiliocannata@gmail.com (A.C.); nferlazzo@unime.it (N.F.); ientile@unime.it (R.I.); moncurro@unime.it (M.C.); giulia.andolina90@virgilio.it (G.A.); 2R & D Regulatory Affairs Department, Medena AG, Affoltern-am-Albis (ZH) CH-8910, Switzerland; chiara.deluca@medena.ch; 3Centre of Innovative Biotechnological Investigations Nanolab (CIBI-NANOLAB), Moscow 119571, Russia; korkina@cibi-nanolab.com

**Keywords:** multiple chemical sensitivity, genetic background role, *ADORA2A* SNP, *ADORA2A* gene expression, PBMC, inflammatory cytokines

## Abstract

Systemic inflammation and immune activation are striking features of multiple chemical sensitivity (MCS). The rs2298383 SNP of *ADORA2A* gene, coding for adenosine receptor type 2A (A2AR), has been involved in aberrant immune activation. Here we aimed to assess the prevalence of this SNP in 279 MCS patients and 238 healthy subjects, and its influence on *ADORA2A*, *IFNG* and *IL4* transcript amounts in peripheral blood mononuclear cells of randomly selected patients (*n* = 70) and controls (*n* = 66) having different *ADORA2A* genotypes. The *ADORA2A* rs2298383 TT mutated genotype, significantly more frequent in MCS patients than in controls, was associated with a three-fold increased risk for MCS (O.R. = 2.86; C.I. 95% 1.99–4.12, *p* < 0.0001), while the CT genotype, highly prevalent among controls, resulted to be protective (O.R. = 0.33; C.I. 95% 0.224–0.475, *p* < 0.0001). Notably, *ADORA2A* mRNA levels were significantly lower, while *IFNG*, but not *IL4*, mRNA levels were significantly higher in TT MCS patients compared with controls. A significant negative correlation was found between *ADORA2A* and both *IFNG* and *IL4*, while a significant positive correlation was found between *IFNG* and *IL4*. These findings suggest that A2AR defective signaling may play a relevant role in PBMC shift towards a pro-inflammatory phenotype in MCS patients.

## 1. Introduction

Multiple chemical sensitivity (MCS), also called environmental sensitivity illness (ESI) or toxicant-induced loss of tolerance (TILT), involves an aberrant susceptibility response to a broad range of chemical substances present in daily life [1]. In the last decades, much attention has been paid to this disorder because of the potential pathogenic role of increased pollution and stressful lifestyles. MCS presents with systemic manifestations, i.e., fatigue, headache, skin rash, breathing problems, diarrhea, bloating, nausea, congestion, itching, sneezing, sore throat, chest pain, changes in heart rhythm, muscle pain or stiffness, dizziness, trouble concentrating, memory problems, and mood changes, that are triggered by acute high-dose exposure or chronic low dose-exposure to physico-chemical factors, such as xenobiotics, heavy metals, radiations, iatrogenic factors, and biological factors, i.e., microbial and food allergens [2,3]. In some countries MCS has also been linked to prolonged exposure to indoor air molds [4]. Disease onset occurs especially in adult life and in women, but the frequency of pediatric cases is increasing [5], and a possible role of in-utero sensitization has been proposed [6].

Despite the absence of validated diagnostic biomarkers, the epidemiological evidence has led the individual countries to at least partially recognize MCS as a pathological state. In Europe, in particular, Germany and Austria classified MCS under the ICD-10 code T78.4 (unspecified allergies, Nitrous Oxide System-hypersensitivity, NOS-idiosyncrasy) [1], while in Japan, where particular attention is paid to environmental pollution, MCS is classified under ICD code T65.9 (unspecified respiratory conditions due to inhalation of fumes, gas, and chemical vapors) [7]. In US and Australia several medical associations have long recognized chemical hypersensitivity as a disability that deserves thorough investigations.

To date the proposed etiopathogenetic mechanism involves an aberrant activation of the vicious cycle N-methyl-d-aspartate (NMDA)-nitric oxide/peroxynitrite triggered by various chemical agents, that ultimately lead to oxidative stress and activation of pro-inflammatory transcription factors [8]. Indeed, dramatically increased levels of reactive oxygen and nitrogen species (ROS, RNS), as well as pro-inflammatory cytokines and the presence of autoimmune antibodies, have been reported in MCS patients compared with healthy subjects [9,10,11,12,13,14]. Moreover, a link has been suggested between MCS and inherited or acquired defects in genes coding for enzymes involved in xenobiotic metabolism phase I and II, antioxidant defense, lipid metabolism and one-carbon pathway [11,12,15,16,17,18,19,20,21]. The presence of polymorphic variants of xenobiotic-metabolizing enzymes, other than raising the risk of serious adverse reactions to drug treatment, can also increase the individual sensitivity to the environmental toxic burden, determining the development of chronic systemic oxidative stress and inflammation.

The release of pro-inflammatory cytokines, as well as the activation and proliferation of T cells, can be inhibited by adenosine receptor activation. Adenosine receptor 2A (A2AR), one of four adenosine receptor sub-types, is present in almost all immune cells, including lymphocytes, monocytes, macrophages and dendritic cells, and its activation increases the production of anti-inflammatory cytokines [22]. The single nucleotide polymorphism (SNP) rs2298383 of *ADORA2A* gene, encoding for A2AR, is a functional variant that may affect the rate of gene transcription due to its location within a regulatory sequence of the gene [23,24]. Notably, recent literature data demonstrate that a reduction of A2AR protein amounts increases the rate of inflammation [25].

We here aimed to assess the prevalence of ADORA2A rs2298383 polymorphism in MCS patients as well as age- and gender-matched healthy subjects, in order to establish a possible association of this SNP with MCS syndrome. Moreover, we evaluated the influence of the rs2298383 SNP on the transcription levels of *ADORA2A* gene, and the consequent effects on the pro-inflammatory phenotype shift of peripheral blood mononuclear cells (PBMC) isolated from both MCS patients and healthy controls.

## 2. Results

### 2.1. Analysis of ADORA2A rs2298383 SNP Prevalence

Both patients and controls were genotyped for *ADORA2A* rs2298383 SNP, and results are shown in Table 1. Genotype distributions in the control group were in concordance with the Hardy–Weinberg equilibrium (HWE) (*p* = 0.928), and with those reported for Italian population in the 1000 Genomes project (http://asia.ensembl.org/Homo_sapiens/Variation/Population?db=core), while they were found to deviate from the expected value by HWE (*p* = 0.000000) in the group of MCS patients. The T mutated allele was more frequent among MCS patients than among healthy subjects, and was predominant in homozygous state, while in controls it was more frequently found in heterozygous state (Table 1). The mutated TT genotype was significantly more frequent in MCS patients than in controls (Table 1). The wild-type CC genotype had a similar prevalence among MCS patients and healthy subjects.

### 2.2. Estimation of Odds Ratios of ADORA rs2298383 Variant Alleles

The mutated TT genotype resulted to be associated with MCS disorder, since the Odds Ratio calculation showed that the risk of developing MCS syndrome increases by about three folds in individuals who have this genotype (O.R. = 2.86; C.I. 95% 1.99–4.12, *p* < 0.0001). Instead, a negative association with MCS was observed for the heterozygous CT genotype, that was significantly more frequent among healthy subjects than in MCS group. Individuals that were carriers of this genotype had a three fold-decreased risk for MCS (O.R.= 0.33; C.I. 95% 0.224–0.475, *p* < 0.0001).

### 2.3. Analysis of ADORA2A Gene Expression

We also investigated the influence of the SNP rs2298383 on *ADORA2A* gene expression levels in 70 MCS patients and 66 healthy subjects randomly selected from the different subgroups representative of the three *ADORA2A* rs2298383 genotypes.

The analysis of gene expression showed that the mean levels of *ADORA2A* mRNA were significantly lower in MCS patients than in healthy subjects (Table 2). Notably, the presence of *ADORA2A* rs2298383 SNP in homozygous state was associated with a decrease in *ADORA2A* mRNA amount both in MCS patients and in controls (Figure 1).

MCS patients with the mutated TT genotype had the lowest *ADORA2A* mRNA expression levels. Significant differences were found in comparison with controls having the same or different *ADORA2A* rs2298383 genotype (MCS TT vs CTR TT *p* < 0.001, vs CTR CT *p* < 0.001, vs CTR CC *p* <0.01), and in comparison with MCS patients having other genotypes (MCS TT vs MCS CC *p* < 0.05, vs MCS CT *p* < 0.01). Significant differences were also found when MCS having either CC or CT genotype were compared with healthy subjects carriers of the same genotypes (*p* < 0.01, *p* < 0.001, respectively). No statistically significant differences were found among controls having different *ADORA2A* genotypes (Figure 1).

### 2.4. Analysis of IFN-γ and IL-4 mRNA and Protein Levels

Given the involvement of *ADORA2A* signaling activation in the modulation of the immune activation, we assessed the expression levels of *IFNG* and *IL4* genes coding for the two inflammatory cytokines IFN-γ and IL-4, linked to the activation of Th1 and Th2 subsets of lymphocytes, respectively, in the same subgroup of patients and controls.

The mean levels of *IFNG*, but not *IL4*, mRNA were significantly higher in MCS patients than in healthy subjects (Table 2). Interestingly, the *ADORA2A* rs2298383 TT mutated genotype was associated with an increase of *IFNG* and *IL4* mRNA levels both in controls and in MCS patients. These latter had the highest amounts of *IFNG* and *IL4* mRNA (Figure 2 and Figure 3).

The *IFNG* mRNA levels in MCS TT individuals were significantly higher than those of healthy subjects (MCS TT vs CTR CC *p* < 0.01, vs CTR CT *p* < 0.001, vs CTR TT *p* < 0.001), while no statistically significant differences were found in comparison with MCS patients having other genotypes. No statistically significant differences in *IFNG* mRNA levels were found when comparing the different genotypes within the healthy cohort. Interestingly, statistically significant differences were observed between either CC or CT MCS subjects and their healthy counterpart (MCS CC vs CTR CC *p* < 0.01, MCS CT vs CTR CT *p* < 0.001).

No statistically significant differences in *IL4* mRNA levels were observed between MCS cases and controls or between different genotype subgroups within the two cohorts (Figure 3).

The assessment of secreted cytokine amounts showed that the mean IFN-γ serum concentrations were three-fold increased in MCS patients compared with healthy subjects (651.5 ± 441.9 vs 202.5 ± 19.14 pg/mL, *p* < 0.0001), while IL-4 concentrations were 1.5 fold-increased in MCS cases compared with controls (6.4 ± 3.8 vs 4.1 ± 1.02 pg/mL, not significant). However, only IFN-γ protein levels resulted to be significantly different between the two groups, likely due to the small number of subjects examined.

Notably, the between-groups comparison of MCS having different *ADORA2A* rs2298383 genotypes showed that carriers of TT mutated genotype had the highest serum levels of IFN-γ and IL-4 (Table 3). However, no significant differences were found when comparing individuals with different genotypes.

### 2.5. Correlation Analysis

Notably, a significant negative correlation was found between *ADORA2A* and both *IFNG* (r = −0.364; *p* = 0.0267) and *IL4* (r = −0.527; *p* = 0.0296) mRNA expression levels, while a significant positive correlation was found between *IFNG* and *IL4* mRNA expression levels (r = 0.724, *p* < 0.0001).

## 3. Discussion

In the last decade, literature data provided evidence for the association of MCS with increased oxidative/nitrosative stress and inflammation [9,10,11,12,13,14], likely resulting from defects in xenobiotic metabolism and antioxidant enzyme defenses [26]. A chronic imbalance in redox homeostasis promotes the development of inflammation mainly through the dysregulation of immune response cells and the activation of pro-inflammatory transcription factors [27]. A key role in oxidative stress-mediated development of inflammatory conditions is played by the loss of adenosine receptor-mediated modulation on immune cells [28]. In particular, adenosine binding to A2AR inhibits the activation, proliferation and commitment of T cells to Th1, as well as the production of pro-inflammatory cytokines, and stimulates, in parallel, the release of anti-inflammatory cytokines [29].

In this study, we first investigated the prevalence of *ADORA2A* rs2298383 SNP, that has been shown to affect A2AR signaling [24,30], in Caucasian MCS patients and healthy subjects. Notably, we here first demonstrated that the *ADORA2A* rs2298383 TT mutated genotype, being largely prevalent in MCS patients, is associated with MCS and represents an unfavorable genetic background for this disorder. Instead, the heterozygous CT genotype, more frequent among healthy subjects, is negatively associated with MCS and displays a protective effect due to a three-fold decreased disease risk. The *ADORA2A* rs2298383 genotype distributions observed in our control group agree with those previously reported for Caucasian population [30], and for Italian general population as reported in our earlier paper [31] and in 1000 Genomes Project (http://asia.ensembl.org/Homo_sapiens/Variation/Population?db=core), an optimal reference population which contains allelic frequencies for a sample of 107 subjects from Tuscany, Italy.

Also interestingly, the TT genotype was associated with a dramatic reduction in *ADORA2A* transcript levels in MCS cases. These findings are apparently in contrast with previously made assumptions. Indeed, on the basis of in silico analysis it has been hypothesized that the wild-type CC genotype is associated with a drastic decrease in *ADORA2A* transcription levels and amount of A2AR molecules, with functional consequences comparable to the receptor blocking by antagonists [24,32]. However, Shinohara and colleagues found that individuals carriers of the *ADORA2A* haplotype including the rs2298383 CC genotype, showed higher amounts of *ADORA2A* mRNA and A2AR protein compared with those having different haplotypes [33]. The present results are in line with those reported by Shinohara and colleagues [33], and suggest that the decrease of A2A receptor levels can lead to a reduced activation of the adenosine signaling pathway. This may result, in turn, in an exacerbated pro-inflammatory response and immune activation in the presence of environmental inflammatory stimuli, thus contributing to MCS symptoms.

Interestingly, given that A2AR is abundantly present in olfactory bulb [34], it is possible to hypothesize that a defective activation of A2AR signaling pathway, caused by genetic defects, may play a role in smell alterations that are reported by MCS patients. Indeed, it has recently been shown that olfactory complaints in MCS patients rely on a complex interaction between genetic and acquired factors, mainly represented by toxic environmental stimuli [35].

We here also first provided a confirmation of this hypothesis by showing that mRNA levels of *IFNG* and *IL4* were higher in subjects carriers of the *ADORA2A* rs2298383 TT mutated genotype than in subjects having either CC or CT genotype. Moreover, the highest serum concentrations of these cytokines were found in TT MCS patients that also had the lowest amount of *ADORA* mRNA transcripts. These observations are supported by the significant negative correlation found between *ADORA2A* mRNA and both *IFNG* and *IL4*. Our data partially replicated the findings of Dantoft et al. [10], that demonstrated higher plasma concentrations of several cytokines including IL-4, but not IFN-γ, in MCS cases than in healthy controls.

IFN-γ plays a key role in macrophage activation, inflammation, and host defense against intracellular pathogens. The inhibition of expression and function of anti-inflammatory molecules represents a key mechanism of IFN-γ-mediated priming of enhanced innate immune responses [36]. The high IFN-γ levels in our MCS cohort indicate a possible role for the deregulation of Th1 subset of lymphocytes in the control of disease-associated inflammation and autoimmunity, and also suggest an increased activation of the immune response against viral infections. Interestingly, some authors have recently advanced the hypothesis of a viral involvement in the pathogenesis of Fibromyalgia and Chronic Fatigue Syndrome, which share many pathognomonic features with MCS [37]. Interestingly, the activation of the interferon pathway has been shown to result in the upregulation of protein kinase-RNA activated (PKR) and 2′, 5′-oligoadenylate synthetase (OAS), and consequently, of RNase-L, leading to RNA interference. This latter, in turn, can reduce mRNA levels and even in some circumstances silence some genes in macrophages [38,39]. The occurrence of this mechanism cannot be excluded in the observed reduction of *ADORA2A* expression in PBMC of MCS patients.

IL-4 is a cytokine associated with the activation of Th2 cells, and has long been considered as one of the tolerogenic cytokines in many autoimmune animal models and clinical settings; however, its overproduction has mainly been associated with the development of allergic reactions and atopy [40]. Notably, MCS patients frequently report co-morbidities, such as hay fever, asthma and atopic dermatitis [3,10]. Moreover, IL-4 plays a role in antagonizing pathogenic Th1 responses [40]. The increased IL-4 serum levels observed in MCS patients recruited for this study and in other studies [9,10], and the here observed significant positive correlation with IFN-γ levels, suggest a compensative reaction to buffer the increase of IFN-γ production.

It is well known that adenosine effects are reduced by caffeine and its metabolites, acting as competitive inhibitors at adenosine receptors throughout the body [41]. Individuals with the *ADORA2A* homozygous mutated genotypes were shown to have lower habitual caffeine consumption compared with wild-type ones, likely due to a mechanism of negative feedback [30,31,42,43]. The high prevalence of *ADORA2A* heterozygous genotype in the general Caucasian population supports the hypothesis that this genetic trait was selected as a result of adaptive evolution to the daily intake of caffeine, the excess of which may display well known toxic side effects for the body. Thus, it is possible to speculate that the *ADORA2A* rs2298383 TT mutated genotype has been subjected to positive selection because it is protective against the long lasting effects of caffeine resulting from its slowed metabolism. A side effect of this natural selection is the increased susceptibility to the onset of inflammatory processes and dysregulation of immune system, as observed in MCS patients that have a defective metabolism of xenobiotics.

These observations suggest that recommendations of family practitioners to MCS patients should include drastic reduction or even elimination of caffeine, that has been shown to reduce *ADORA2A* mRNA expression in a dose- and time-dependent manner [44]. Moreover, the administration of molecules capable of targeting A2AR, such as the non selective 5′-N-ethylcarboxamine (NECA) and the selective CGS-21680 (2-phenethylamino-substituted NECA) and PSB-0777 (2-phenylsulfonate-substituted NECA), already developed and tested in inflammatory disorders and other disease paradigms [25,28], could be regarded as an effective pharmacological approach for patients with severe MCS.

Our preliminary findings suggest the important role of A2AR signaling in the chronic inflammatory process associated with MCS. Further replication studies will be useful to confirm the present observations.

## 4. Materials and Methods

### 4.1. Editorial Policies and Ethical Considerations

This study conforms with the Declaration of Helsinki (amend. 2013). All subjects recruited for this study provided written informed consent to the collection of anamnestic data and peripheral blood sampling in EDTA tubes aimed to carry out research investigations.

### 4.2. Study Cohorts

This study was carried out using a biobank of whole blood samples collected from 279 Italian MCS patients (M = 59, F = 220; 45.6 ± 11.3 years; 78.8% females), and 238 Italian healthy volunteers (M = 63, F = 175; 38.2 ± 12.4 years; 73.5% females). The enrollment of MCS patients was mostly carried out at Department of Medical Pathophysiology, University of Rome “La Sapienza”, Polyclinic Umberto I, and at IDI IRCCS (Istituto Dermopatico dell’Immacolata; study protocol no.121/CE/2008, December 11th 2008)(Rome, Italy).

All subjects, presenting with symptoms compatible with MCS and QEESI score > 20 [2], exhibited biochemical features of high oxidative stress (whole blood chemiluminescence above 500 cps/μL) and inflammation, and had a genetic background suggesting defects in the metabolism of xenobiotics, as reported in previous papers of our group [9,16,18,19,43]. Nonsmokers were 83.1%; alcohol or drug abusers were not present.

Healthy subjects were recruited either among staff members of Polyclinic Umberto I, University “La Sapienza” and IDI IRCCS (study protocol no. 121/CE/2008, 11th December 2008), or among staff members and volunteer blood donors at Polyclinic Hospital University of Messina (study protocols no. 37/17, 10th May 2017, and 51/17, 10th July 2017), according to the following established inclusion criteria: a) absence of any clinically diagnosed disease; b) absence of allergic or immunologic disturbances; c) absence of supplementation with drugs or nutraceutical compounds in the last six weeks, at the time of blood sampling; d) absence of oxidative stress, i.e., whole blood total production of ROS/RNS below 500 cps/μL, as previously reported on the basis of a luminol-dependent chemiluminescent response to phorbol 12-myristate 13-acetate [9]; e) non-smoking habits; f) no alcohol consumption or drug abuse.

### 4.3. Isolation of Genomic DNA

Genomic DNA (gDNA) was isolated from peripheral white blood cells using the Gentra Systems PureGene-DNA purification kit (Qiagen, Milan, Italy), according to manufacturer’s instructions. The DNA was quantified by spectrophotometric measurement at 260 nm using a Biophotometer (Eppendorf AG, Hamburg, Germany). DNA quality was considered acceptable for samples having a 260/280 ratio ≥ 1.6. DNA integrity and the presence of contaminant RNA were checked by electrophoresis on 0.8% agarose gel, and subsequent UV detection of DNA bands using a gel photodocumentation system (Vilber Lourmat, Marne-la-Vallée, France).

### 4.4. Genotyping

The screening for the presence of ADORA2A rs2298383 (C > T) SNP was performed by Real-time PCR-based allelic discrimination using a pre-designed TaqMan-based Genotyping Assay (ID: C_16189248_10; ThermoFisher, Monza, Italy), as previously described [31]. PCR reactions were carried out in a 7900HT Fast Real-Time PCR System (Applied Biosystems, Foster City, CA, USA), using thermal cycling conditions suggested by manufacturer’s protocols.

### 4.5. Gene Expression Analysis

Total RNA was extracted and purified from PBMC of randomly selected MCS patients (*n* = 70; 18M, 52F) and healthy subjects (*n* = 66; 16M, 50F), having different ADORA2A rs2298383 genotypes (MCS: CC = 22, CT = 24, TT = 24, controls: CC = 22, CT = 22, TT = 22). PBMC were isolated from 300 µL of whole blood samples, after red blood cells lysis by RBC buffer available from Gentra Systems PureGene-DNA purification kit (Qiagen, Milan, Italy). The isolated RNA was quantified by spectrophotometric measurement at 260 nm using a Biophotometer (Eppendorf AG, Hamburg, Germany), then, 2 μg RNA were reverse-transcribed to cDNA using the High-Capacity cDNA Archive kit (Thermo Fisher, Monza, Italy) according to manufacturer’s instructions.

mRNA transcript levels were quantified by SYBR green-based Real-time PCR, using β-actin (*BACT*) as endogenous control. PCR reactions were set up in a 96-well plate on a 7900HT Fast Real-Time PCR System (Applied Biosystems, Foster City, CA, USA), and were carried out in a final volume of 20 μL, containing Master mix (2×) Power up SYBR Green PCR (Invitrogen Life Technologies, Milan, Italy), 0.2 µM for each primer, and 40 ng of cDNA. The used primer sequences were 5′-GGCTGCCCCTACACATCATC-3′ (fwd) and 5′-GCCAGGTACATGAGCCAGAGA-3′(rev) for *ADORA2A*, 5′-GCAGCCAACCTAAGCAAGAT-3′ (fwd) and 5′-TCACCTGACACATTCAAGTTC TG-3′ (rev) for *IFNG*, 5′-CCTCTGTTCTTCCTGCTA-3′ (fwd) and 5′-AGATGTCTGTTACGGTC-3′ (rev) for *IL4*, and 5′-TTGTTACAGGAAGTCCCTTGCC-3′ (fwd) and 5′-ATGCTATCACCTCCCCTG TGTG-3′ (rev) for *BACT*. A final dissociation curve analysis was carried out to evaluate the amplification specificity.

Data were analyzed by recording the Ct of target and endogenous genes for each sample, and subsequently calculating the ∆Ct in order to use the 2^−ΔCt^ method for assessing the absolute levels of *ADORA2A*, *IFNG*, and *IL4* mRNA transcripts.

### 4.6. Assessment of IFN-γ and IL-4 Serum Concentrations

The determination of IFN-γ and IL-4 secreted protein levels was carried out in a smaller subgroup of subjects (MCS = 31F, CTR = 16F), due to unavailability of serum samples. In order to minimize gender variations only female patients were included in these analyses. The measurements of cytokine serum levels were performed using a commercially available assay (BioRad, Segrate, Italy) as described by De Luca and co-workers [9].

### 4.7. Statistical Analyses

Continuous data are expressed as means ± standard deviation (S.D.), and the categorical variables as number and percentage. The Fisher’s exact test was applied to test the difference between cases and controls in terms of categorical variables, and the compliance of *ADORA2A* rs2298383 genotype distributions to HWE, based on a Web program (http://ihg.gsf.de/cgi-bin/hw/hwa1.pl).

A sample size analysis was performed to calculate the minimum sample size required for gene expression studies. The sample size was calculatedconsidering *ADORA2A* mRNA levels as the primary outcome variable. Taking into account a power of 80% and an alpha of 0.05, and a number of groups equal to three (as per three different genotypes), it was determined that a minimum number of 69 MCS patients would be needed. Thus, we chose to analyze gene expression levels in 70 MCS subjects.

The distribution of values obtained by gene expression studies was tested by Kolmogorov-Smirnov test, and was not found normal. As a consequence, non-parametric tests were used for subsequent statistical analyses. Differences in the mean mRNA total levels of *ADORA2A*, *IFN*G and *IL4* between MCS group and control group were analysed by Mann-Whitney test, while differences between subgroups having different *ADORA2A* genotypes within either the MCS cohort or healthy cohort were analyzed by Kruskall-Wallis test. A Spearman correlation analysis was performed to test the relationships between the examined continuous variables.

Statistical analyses were performed using the software GraphPad Prism 5 (GraphPad Software, San Diego, CA, USA). A *p*-value ≤ 0.05 was considered statistically significant for all the analyses.

## Figures and Tables

**Figure 1 ijms-21-01858-f001:**
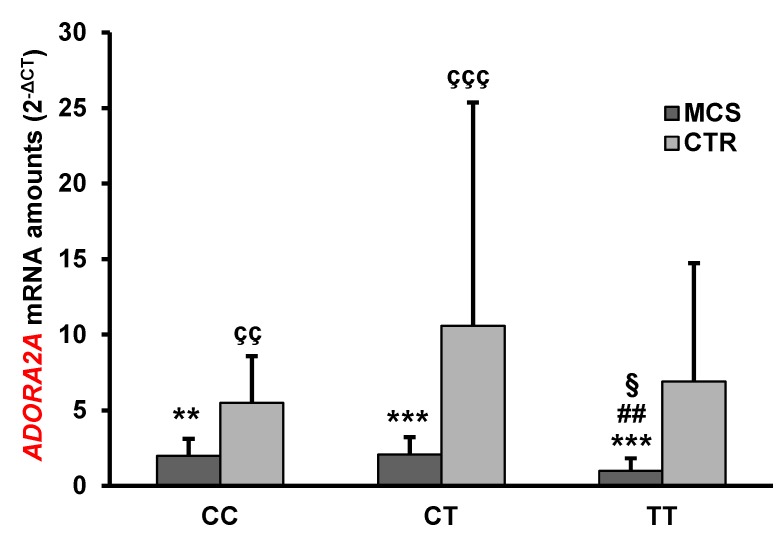
Mean *ADORA2A* mRNA expression levels in PBMC of MCS patients and healthy subjects (CTR) having different *ADORA2A* rs2298383 genotypes. Values shown are means ± SD. ** *p* < 0.01, *** *p* < 0.001 significant difference in comparison with CTR having the same genotype; ^##^
*p* < 0.01 significant difference in comparison with CT MCS; ^§^
*p* < 0.05, significant difference in comparison with CC MCS; ^çç^
*p* < 0.01, ^ççç^
*p* < 0.001 significant difference in comparison with TT MCS.

**Figure 2 ijms-21-01858-f002:**
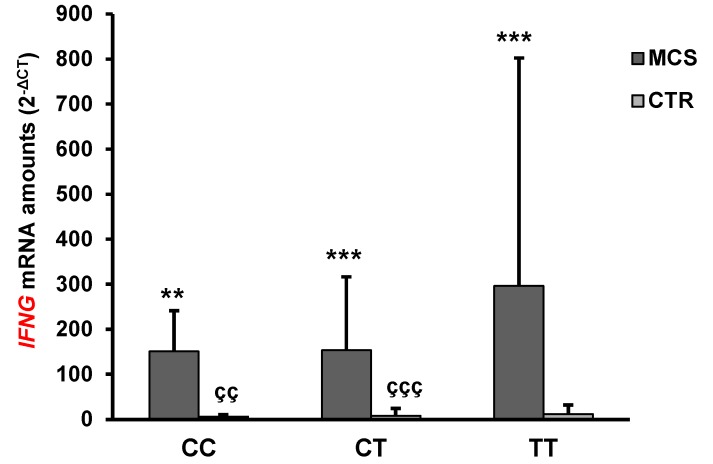
Mean *IFNG* mRNA expression levels in PBMC of MCS patients and healthy subjects (CTR) having different *ADORA2A* rs2298383 genotypes. Values shown are expressed as means ± SD. ***p* < 0.01, *** *p* < 0.001, significant difference in comparison with CTR having same genotype; ^çç^
*p* < 0.01, ^ççç^
*p* < 0.001, significant difference in comparison with TT MCS.

**Figure 3 ijms-21-01858-f003:**
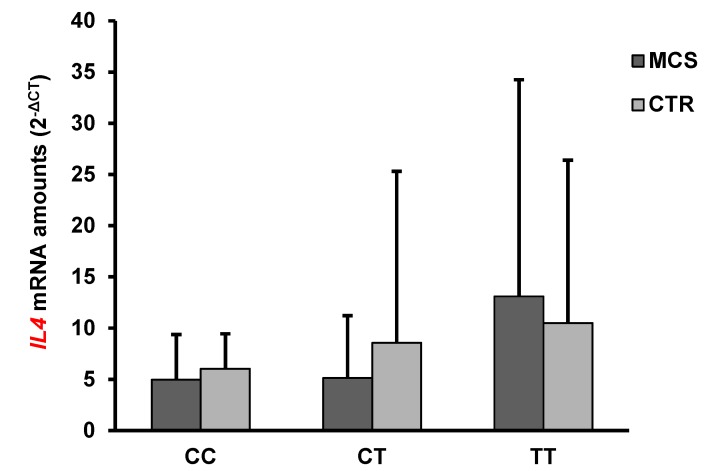
Mean *IL4* mRNA expression levels in PBMC of MCS patients and healthy subjects (CTR) having different *ADORA2A* rs2298383 genotypes. Values shown are expressed as means ± SD.

**Table 1 ijms-21-01858-t001:** Prevalence of *ADORA2A* rs2298383 SNP in MCS patients and healthy subjects.

*ADORA2A* *rs2298383* Genotype	MCS (*n* = 279)	Controls (*n* = 238)	*p*-Value
CC	20.1% (56)	20.2% (48)	1
CT	24% (67)	49.1% (117)	<0.00001
TT	55.9% (156)	30.7% (73)	<0.00001
Mutated allele T frequency	0.679	0.4475	-
Wild-type allele C frequency	0.321	0.5525	-

**Table 2 ijms-21-01858-t002:** mRNA expression levels of genes coding for *ADORA2A* and inflammatory cytokines in PBMC of MCS patients and healthy controls.

mRNA Levels (2^−∆Ct^)	MCS (*n* = 70)	Controls (*n* = 66)	*p*-Value
*ADORA2A*	1.72 ± 1.15 ***	7.66 ± 12.0	<0.0001
IFNG	200.65 ± 315.05 ***	14.9 ± 42.0	<0.0001
*IL4*	7.55 ± 12.72	8.8 ± 14.7	n.s.

Legend: Values shown are expressed as means ± SD. *** *p* < 0.001 significant difference in comparison with controls.

**Table 3 ijms-21-01858-t003:** Serum concentrations of secreted inflammatory cytokines in MCS patients having different *ADORA2A* rs2298383 genotypes.

*ADORA2A* rs2298383 Genotype	IFN-γ (pg/mL) ^§^	IL-4 (pg/mL) ^§^
CC	488.8 ± 364.7	5.5 ± 3.6
CT	515.7 ± 295.0	5.1 ± 3.7
TT	722.8 ± 461.1	6.7 ± 3.5

**^§^** Values shown are expressed as means ± SD.

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
