# Peer review of "The SNP rs2298383 Reduces ADORA2A Gene Transcription and Positively Associates with Cytokine Production by Peripheral Blood Mononuclear Cells in Patients with Multiple Chemical Sensitivity"

_ijms, 2020, doi:10.3390/ijms21051858_

Round 1

Reviewer 1 Report

In the abstract you wrote:

Notably, ADORA2A mRNA amount was significantly lower, while IFN-γ amount, but not IL-4, was significantly higher in TT MCS patients compared with controls. A significant negative correlation was found between ADORA2A and both IFN-γ and IL-4, while a significant positive correlation was found between IFN-γ and IL-4.

It is not clear from the abstract: you speak about mRNA levels or the levels of the cytokine produced? Some read only abstracts. You should say clearly what do you mean. The authors may consult with a more knowledgeable person to estimate whhther the word amount is an appropriate one.

You whore:

These findings suggest that

Row 26: A2AR defective signaling may play a relevant role in PBMC priming towards a pro-inflammatory

phenotype in MCS. I think the wording should be play a role towards pro-inflammatory shift. Priming is when an immune cell encounters an antigen for the first time.

Introduction:

What is head-related? what is head-chest related?

Row 72. that a reduction of A2AR protein amounts increase the rate of inflammation [25]. Should be increases

You wrote: Result obtained by genotyping of patients and controls for ADORA2A rs2298383 SNP are shown in Table 1.

Should be results are shown. From this point I will not pick up some grammar mistakes. The authors are requested to proof read the manuscript by an English speaking specialist.

What is HWE? Should be opened

Row 235 caffeinand should be corrected.

It would be interesting if the author can open what molecules are used to stimulate A2AR

Author Response

We greatly thank the reviewer for the precious comments. The manuscript was revised accordingly.

A point-by-point reply follows.

In the abstract you wrote:

Notably, ADORA2A mRNA amount was significantly lower, while IFN-γ amount, but not IL-4, was significantly higher in TT MCS patients compared with controls. A significant negative correlation was found between ADORA2A and both IFN-γ and IL-4, while a significant positive correlation was found between IFN-γ and IL-4.

It is not clear from the abstract: you speak about mRNA levels or the levels of the cytokine produced? Some read only abstracts. You should say clearly what do you mean.

We clarified by adding “mRNA levels” in the mentioned sentence.

The authors may consult with a more knowledgeable person to estimate whhther the word amount is an appropriate one.

The word amount  was  replaced by “levels”.

You whore:

These findings suggest that

Row 26: A2AR defective signaling may play a relevant role in PBMC priming towards a pro-inflammatory phenotype in MCS. I think the wording should be play a role towards pro-inflammatory shift. Priming is when an immune cell encounters an antigen for the first time.

The sentence was corrected as suggested.

Introduction:

What is head-related? what is head-chest related?

We opened the two mentioned words and chose to clearly mention all common MCS symptoms (revised sentences in rows 37-39).

Row 72. that a reduction of A2AR protein amounts increase the rate of inflammation [25]. Should be increases

The word was corrected as suggested

You wrote: Result obtained by genotyping of patients and controls for ADORA2A rs2298383 SNP are shown in Table 1.

Should be results are shown. From this point I will not pick up some grammar mistakes. The authors are requested to proof read the manuscript by an English speaking specialist.

We corrected the mentioned sentence and proofread the text.

What is HWE? Should be opened

We opened HWE by writing Hardy Weinberg equilibrium in full.

Row 235 caffeinand should be corrected.

Row 235 was corrected as suggested

It would be interesting if the author can open what molecules are used to stimulate A2AR.

We mentioned in the text the commonly used agonists of A2AR, that are the non selective as the non selective 5’-N-ethylcarboxamine (NECA) and the selective CGS-21680 (2-phenethylamino-substituted NECA) and PSB-0777 (2-phenylsulfonate-substituted NECA)(Müller et al. 2020 Methods Mol Biol)

Reviewer 2 Report

A significant improvement over the first paper. There are a few typos still evident. e.g. the use of the plural "folds" instead of fold. A couple of places there is no space evident between two words (e.g. lines 132 and 235). A few grammar issues the proof reader can remove.

Author Response

We greatly thank the reviewer for the time spent on our manuscript, that was modified by correcting wrong typos, the word folds and grammars issues as suggested.

This manuscript is a resubmission of an earlier submission. The following is a list of the peer review reports and author responses from that submission.

Round 1

Reviewer 1 Report

The authors address an interesting issue, that is the presence of genetic factors potentially favoring the development of an inflammatory response and its exacerbation in patients affected by multiple chemical sensitivity syndrome. The work is original and well organized, the methods are scientifically sound and adequately described, the conclusions are supported by the results.

However, some minor criticisms remain to be addressed.

1) Among the co-authors of this paper are De Luca and Korkina, who previously reported on biochemical features of MCS patients, including other cytokines and chemokines that were not examined in the present study. Would the authors provide a rationale for the selection of IFN-γ and IL-4 cytokines among others, and for having included in gene expression studies only 40 and 66 subjects from the group of 279 MCS patients and 239 healthy subjects, respectively, and even less individuals for assessment of serum cytokine levels?   

2) Literature data report that most of MCS patients present with olfactory impairment and subjective olfactory complaints. Would the authors discuss about previous research in the field of smell dysfunction in MCS and add some comments on the potential role of adenosine receptor signaling alterations in the development of these symptoms?

3) In the section Materials and Methods the total number of MCS patients does not correspond to the M+F sum within brackets.

4) In the Graphics shown in Figs 1 and 2, statistically significant differences are not shown; in the Graphics of  Figs 1, 2 and 3 the y axis is not shown.

5) Replace means ± DS by means ± SD in the legends of figures and Table 3.

6) In Table 2 add “Values shown are expressed as means ± SD”.

Reviewer 2 Report

Review of the paper by Cannata et al.

Multiple Chemical Sensitivity is a neglected and denied disease in many countries. Therefore any scientific investigation  about the mechanisms of this  devastating disease is welcome.

The paper of Cannata A et al is interesting. Here please find my comments and suggestions for the improvement.

The title. I would better say that this SNP positively associates with….. I think the word association is much better to use in your paper instead of e.g. modulation or risks. My suggestion is that the title would be: The SNP …… positively associates with cytokine production by peripheral blood mononuclear cells in patients with  Multiple Chemical Sensitivity.  PBMC is an acronym and should not be used in the title because  some  readers may not understand. In this paper there is no assessment of risks as e.g. calculation of RR or OR ( at least they are not mentioned in the results), therefore I suggest to correct  throughout the text  all expressions that authors would mean as association.  If you calculated ODs - please make a special paragraph and  show the data.   The abstract The authors say that the SNP   increases the risk by three fold. If you want to say about the risks then you should   convince readers by giving the OR and the p-values. You say about protective effect. I would  use the wording negative association, there should be p value after the OR value to show how significant is this association if you show OR. In the introduction you may mention that in some countries MCS has been linked to prolonged exposure to indoor air molds. There are publications on that issue. row 52 NMDA- please open  the acronym row 65 Is it correct to say that the receptor increases the production of cytokines? Maybe you should say that the activation of the receptor….. row 71 and in the text throughout: Is the better wording instead of distribution to use the word prevalence? row 100 and throughout the text. Is the word level better than the word amount? Footnote to Fig 1: DS  is SD ( standard deviation). This mistake was found  elsewhere ,  please check and correct. Table 3: were the levels of the cytokines significantly different in  different phenotypes?  It seems – not. It would be interesting to have these data in the table 3. Discussion is too lengthy. I suggest to shorten it and discuss the points that are relevant only to the results. There is too much basic immunology, too much discussion of the effects of caffeine. Unfortunately, too lengthy discussion will take the attention away from the results of the study. I found some sentences cumbersome. Again, here we should speak only about association. Material and methods:

Study cohorts ,  not cohort.

You claim that the MCS  patients were selected by QEESI   questionnaire and you say  that they exhibit futures of  oxidative stress and inflammation. Where are the data on inflammation in the MCS patients?

You present ROS/RNS values for the controls. What method has been used?  What are the data for the MCS?  What is the cut-off in the method? I am very interested in these data. Can it be used to discriminate patients from controls?

Preparation of PBMC is not described.

You may briefly mention how the INFg and IL-4 were measured, although  it has been described earlier.

In conclusion:  My suggestion is to improve the description of methods (patient selection) and to reduce discussion  to approximately one third of the present. In the discussion you should estimate what practical value this finding will have to medical doctors.

Reviewer 3 Report

RS2298383 ADORA2A GENE.

Patient selection and comparison of frequency data.

The MCS and Controls were selected from a databank of samples. The controls were selected in a hospital environment and were selected on a specific basis, those with MCS and those considered healthy. Why were these control people sampled in the first place?

We do not know the demographics of the population % females, age etc. We are not told of the ethnicity of the test or control populations. This type of demographic data should be shown. The demographic data should also be shown for the sub-sample tested for the cytokines. A sample size analysis could be used to determine the sample size for the subgroups.

Bases

Control

MCS

Pop Norm

1000 genomes

CC

.202

.201

.187

CT

.491

.240

.458

TT

.307

.559

.355

Table 1 should be compared to the 1000 genomes distribution of Toscani in Italy as that does give the population norm which will allow assessment of the validity of both the test and control comparison datasets. What we do see is the high similarity between the control dataset and the population norms for Italians which is good. (http://asia.ensembl.org/Homo_sapiens/Variation/Population?db=core;r=22:24419543-24439542;v=rs2298383;vdb=variation;vf=210359408 ). What we are seeing is a shift toward a homozygote status for ADORA2A. So, what we should see in the MCS is a similar alteration of the ADORA2A dependent levels of measures.

In table 2 and figure 1 we are shown the levels of ADORA2A mRNA, IFN-gamma and IL-4. It is ADORA2A mRNA and INF-gamma that are elevated. The problem I have with figure 1 is that the controls with the TT SNP set have significantly elevated ADORA2A mRNA compared with the MCS TT subgroup. This surely indicates that the ADORA2A SNP mutation is unlikely to be the cause of the variation as the TT controls should have similar relationships to the TT MCS. A correlation analysis was performed between the continuous variables and IFN-gamma and IL4, were reported. Why was there no reporting of the association between ADORA2A mRNA levels and IFN-gamma and IL4? This would be required to establish a positive relationship between ADORA2A and IFN-gamma. If this is positive and significant then you may have a publishable paper based upon the data. If it is not than one would need to look for other reasons for the increased IFN-gamma and to discuss these in the discussion.

Activation of the interferon pathway can occur via multiple different mechanisms, from viral double stranded RNA through to multiple xenobiotic chemicals e.g. benzene. The chemicals to which the patients are reacting. There is insufficient evidence in this presented dataset to allow one to draw the conclusion that the mutations in ADORA2A (rs2298383) in the homozygote state result in activation of Interferon. Activation of the interferon pathway will result in upregulation of PKR and 2’5’-Oligoadenylate synthetase. The resultant upregulation of RNase-L will induce an effect termed “RNA interference” which can reduce mRNA and even in some circumstances silence some genes. This mechanism has not even been discussed in the discussion. I would search this mechanism as a priority as what we may be observing here is the effect of RNA interference driven by upregulation of the interferon pathway.